# Effect of the Maxillary Sinus on Tooth Movement during Orthodontics Based on Biomechanical Responses of Periodontal Ligaments

Xin Liu [1],[†], Mao Liu [2,3,4],[†], Bin Wu [5], Jingjing Liu [2,3,4], Wencheng Tang [1],*and Bin Yan [2,3,4],*

1   School of Mechanical Engineering, Southeast University, Nanjing 211189, China; 101000185@seu.edu.cn
2   Department of Orthodontics, The Affiliated Stomatology Hospital of Nanjing Medical University, Nanjing 210029, China
3   Jiangsu Province Key Laboratory of Oral Diseases, Nanjing 210029, China
4   Jiangsu Province Engineering Research Center of Stomatological Translational Medicine, Nanjing 210029, China; maoliu@stu.njmu.edu.cn (M.L.); liujingjing_ortho@njmu.edu.cn (J.L.)
5   College of Mechanical and Electronic Engineering, Nanjing Forestry University, Xuanwu District, Nanjing 210037, China; wubin@njfu.edu.cn
*   Correspondence: 230159314@seu.edu.cn (W.T.); byan@njmu.edu.cn (B.Y.)
†   These authors contributed equally to this study.

**Abstract:** The maxillary sinus is a common anatomic limitation for orthodontic tooth movement. The effect of orthodontic forces on a particular anatomy can be studied using finite element analysis (FEA). Our study aimed to determine the effect of different tooth penetration depths into the maxillary sinus floor (MSF) on the orthodontic force system for bodily tooth movement. Using the cone-beam computed tomography of a patient with low MSF, we modeled the geometry of canine, premolar, and molar teeth with their periodontal ligaments and the alveolar bone surrounding them. The models were manually modified to simulate different root penetration depths. Thereafter, the center of resistance and stress distributions for teeth penetrating into the MS were determined using FEA. Moreover, the force systems for teeth with a low MSF to varying degree were evaluated based on the FEA results. During orthodontic tooth movement, the individual differences in the periodontal anatomy should be considered. The CR position decreases with the penetration depth, while the average hydrostatic stress in the PDL increases rapidly. In this paper, we present the correction coefficients of the orthodontic force and moment for a tooth penetrating into the MSF, which is necessary for personalized treatment planning.

**Keywords:** maxillary sinus; tooth movement; center of resistance; periodontal ligament; numerical simulation





## 1. Introduction

Orthodontics generally deal with the correction of misaligned teeth under the application of orthodontic forces. Tooth movement triggered by orthodontic force is affected by the biomechanics properties and anatomy of teeth and the periodontal tissue. The maxillary sinus (MS) is one of the anatomical limitations during orthodontic treatment.

The MS is an air-filled cavity lined with mucosa between the floor and the posterior teeth [1]. It may extend laterally and inferiorly due to growth, absence of teeth, or other factors. For around 50% of adults, the maxillary sinus floor (MSF) extends close to the maxillary posterior teeth [2]. The root apices of premolar, molar, and canine teeth can penetrate into the MS in some cases [3–7] and lose the support of the alveolar bone. The MSF is a hard high-density bone similar to the cortical bone [8]. Bone remodeling for the MSF is more difficult than that for the cancellous bone. These factors generally affect orthodontic tooth movement (OTM) and treatment outcome.

For patients with a low MSF, teeth moving through the MS (TMTMS) becomes inevitable in some cases, such as in canine distalization [9] or when the spaces closes after a tooth extraction [10]. Bodily movement is generally expected but is difficult to achieve in these cases. When roots penetrate into the MS, moderate apical root resorption [11], considerable tipping [12], abnormal pulp vitality, or perforation of the sinus membrane [13] can be observed during OTM. Careful modification of the orthodontic force system for the tooth penetrating into MSF is helpful in reducing these side effects [8].

Several successful practices of TMTMS have been reported in recent years [14,15]. On the basis of clinical experience, it is universally recognized that light constant force can move teeth through the sinus floor [12]. However, few researchers have addressed the effect of MS anatomy on tooth movement in more detail.

Finite element analysis (FEA), which has long been used in biomechanics to study the responses of complex anatomies under specific loads, has also been widely exploited in orthodontics. One advantage of FEA is the controllable and limitless experimental conditions. In this study, we aimed to assess the biomechanical response of teeth moving through the MSF at different penetration depths. The center of resistance (CR) and periodontal stress distributions of teeth penetrating to different depths were evaluated using the finite element method (FEM). The orthodontic force and moment-to-force ratio (M/F) for bodily movements were evaluated based on the FEA results. Our findings contribute to a better understanding of the mechanisms in teeth moving through the MS.

## 2. Materials and Methods

3D models were constructed from tomography images. We used the cone-beam computed tomography (CBCT) image of a patient with low MSF (Figure 1) from the Affiliation 2. The tomography was scanned using a NewTom 5G (version FP, Imola, Bologna, Italia) under the following settings: mode (81.5 mGy·cm, 110 KVP, and 5 mA); voxel size, 0.25 mm; and slice thickness, 0.25 mm. The scan was originally taken for diagnosis before orthodontic treatment.

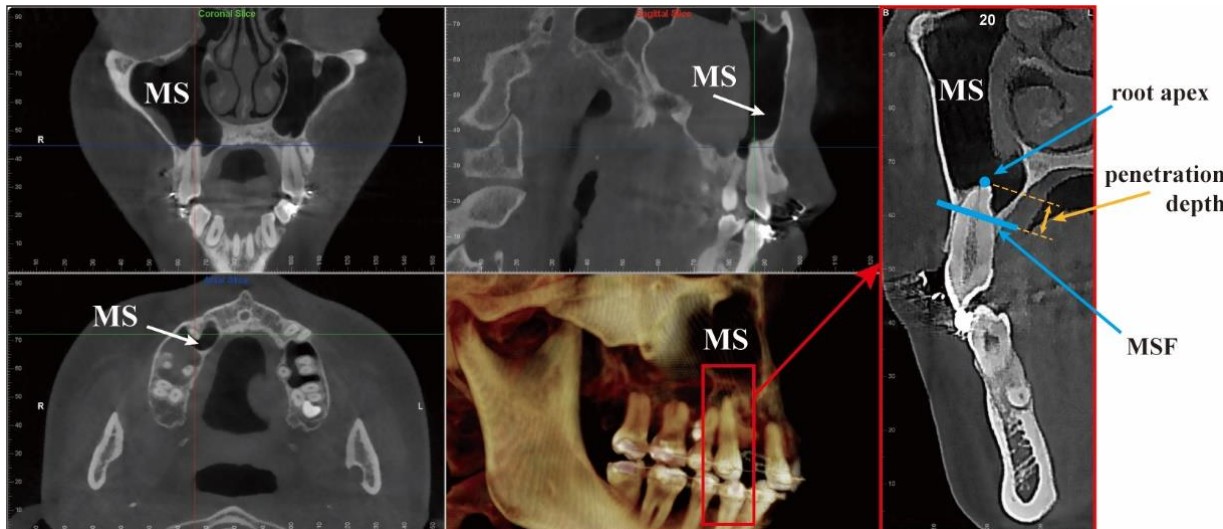

**Figure 1.** Tomography images of a patient with a low MSF. Penetration depth is the distance between the root apex and the MSF; MS is the maxillary sinus; MSF is the maxillary sinus floor.

Three types of teeth were analyzed, including a canine, a double-rooted premolar, and a molar with three roots. The lengths of the teeth (i.e., the long axis) were 23.41 mm, 19.70 mm, and 19.43 mm, respectively. Their shapes and sizes are shown in Figure 2.

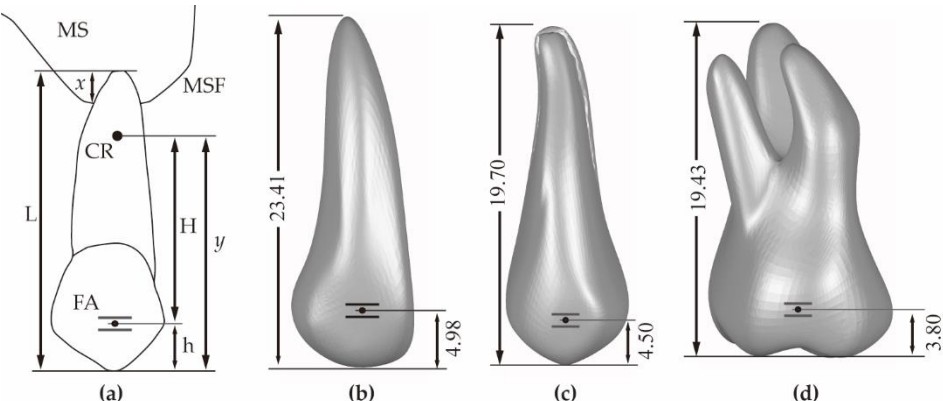

**Figure 2.** Schematic of critical dimension variables (**a**) and the models of the canine (**b**), the premolar (**c**), and the molar (**d**). CR: the center of resistance of the tooth; FA: the facial-axis point on the crown; L: the length of the teeth; h: the bracket height, the distance between FA and the tooth cusp or edge; H: the distance between CR and FA; $x$: penetration depth of the root; $y$: the indicator of CR position, the distance from CR to the tooth cusp or edge.

The tomography images of teeth, PDL, and the alveolar bone were segmented by defining the thresholds of the greyscale values in Mimics (version 19.0; Materialise, Leuven, Belgium). The model comprised teeth, a 0.2 mm PDL, and the manually adjusted alveolar bone. The surface models constructed in Mimics were imported into Geomagic Wrap (version 2017; Geomagic, Raleigh, NC, USA) for constructing the geometry entities. After the laborious work of model processing and repair, which included removing model noise and repairing small holes and smooth and sharp edges, the geometric models were completed, as shown in Figure 3.

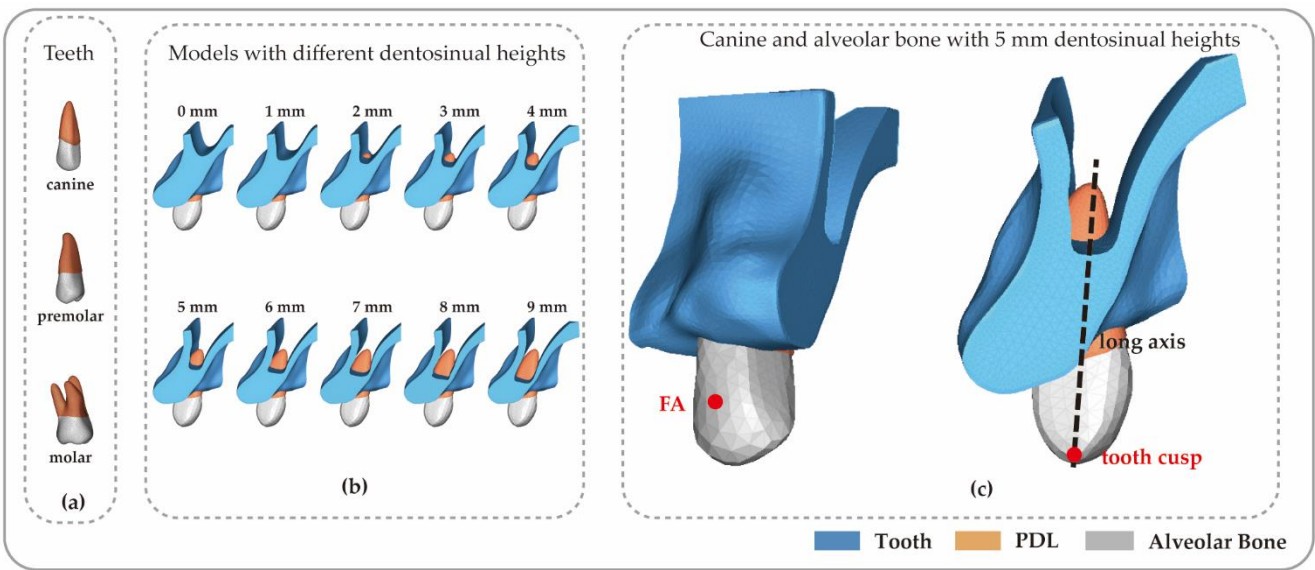

**Figure 3.** Finite element models with 10 different penetration depths for each of the three teeth. (**a**) Finite element models of three types of teeth. (**b**) The manually modified alveolar bone models for the canine. (**c**) The model of the canine and its alveolar bone with 5 mm depth of penetration. FA is the facial-axis point on the crown. (Boundary condition, fixed).

To analyze the influence of the MS anatomy, the penetration depth was individually adjusted for each subject by lowering the MSF. The cavity above the alveolar bone is the MS and there is no material filling in the models. The MS floor is the cortical bone between the cancellous bone and the MS. The roots of teeth penetrated into MS to different depths because of the various heights of the MSF. Penetration depth ($x$) is the portion of the root

penetrating into the MS. It is difficult to obtain real cases involving various depths in clinical practice, which is the advantage of FEA. Various distances between root apices and the MSF were established by modifying the alveolar bone manually (Figure 3b). In the models of different subjects, the depths varied from 0 to 9 mm incrementing 1 mm. Therefore, we had 10 subjects for each of the three types of teeth as shown in Figure 3. The penetration depth was the only difference between subjects with the same tooth.

The geometric entities were imported into Abaqus (version 2018; Dassault System, Concord, MA, USA) for FEA. Orthodontists generally stick the bracket to the crown at the facial-axis (FA) point [16]. Therefore, the orthodontic force in this study was applied on the FA point of the crown. The boundary condition was defined by fixing the top and lateral surface of the bone to reflect the natural intraoral environment. The bodily tooth movement in the mesiodistal direction was simulated in this study.

To develop a simplified model for reducing the analysis time, the alveolar bone and the teeth were considered to be linearly elastic, homogeneous, and isotropic, [17] as summarized in Table 1. The non-linear properties of PDL are generally recognized [18]. Furthermore, the incorporation of the nonlinearity of the PDL can significantly alter the FE analysis [19]. In this study, the PDL was assumed to be visco-hyperelastic material [18]. The constitutive model was implemented into Abaqus by a user-defined material subroutine. The material properties of these components are presented in Table 1.

**Table 1.** Material properties.

| Material | Young's Modulus (MPa) | Poisson's Ratio |
|---|---|---|
| Alveolar Bone | 490 | 0.30 |
| Tooth | 18,600 | 0.31 |
| PDL | Viscoelastic | 0.45 |

The center of resistance (CR) is a critical reference point for orthodontic treatment. Orthodontists generally expect the force to act through that point, so that the tooth moves without rotation (i.e., pure translation) [20]. The stress along the PDL is another important factor affecting tooth movement. Ignoring the friction of the appliance, the stress was assumed to be the indicator of the orthodontic force moving the tooth [17]. Then, the magnitude of force was evaluated by analyzing the hydrostatic stress level for the bodily movement. Therefore, we suggest two methods for evaluating the CR position and the periodontal stress level of the teeth under the influence of the MSF sinking.

*2.1. Computing the CR Position Using the Virtual Force Value Method*

When the tooth is in pure translation under a horizontal force, the CR is the acting point through which the force passes [17]. The CR estimation should be based on the morphological variation of teeth and periodontal tissues [21].

During orthodontics, two principal forces are applied to the tooth: the orthodontic force and the reaction force from the periodontium. They always act in two opposite directions. If the orthodontic force has a common line of action with the reaction force, the tooth generates a bodily movement, and the acting point of the force is the CR of the tooth. If the forces act on the tooth at an offset, they will generate a tipping movement. This is a more common phenomenon in clinical practice since the orthodontic force is applied at the bracket. However, an additional pure moment could move the individual force off its line of action from the bracket to the CR. Therefore, bodily movement can be produced with orthodontic force coupled with a particular moment. The relationship between the value of the moment $M$ and the offset distance (H) from the bracket to the CR can be directly described with Equation (1) [22].

$$M = F \times H \tag{1}$$

where *F* is the value of the force acting on the facial-axis point (FA) (or the bracket), *M* is the value of the additional moment offset the force *F*, and H is the distance between CR and FA.

Therefore, the CR position could be indicated by the distance from the CR to the tooth cusp:

$$y = \mathrm{H} + \mathrm{h} = \frac{M}{F} + \mathrm{h} \tag{2}$$

where *y* is the distance between CR and the cusp, H is the distance between CR and FA point, and h is the offset of FA from the tooth cusp.

A virtual force value method was used in this study to evaluate the CR position of the specific tooth based on the relationship between the CR position and the M/F for bodily movement in Equation (1). The aim of the method is to calculate the offset of the CR from the cusp (*y*) using the FEM. The force system of the tooth in this study was applied on the FA point. The position of the FA point is shown in Figure 2.

A constant force and a changing moment were applied to the tooth model. The force was set as 1.0 N. The moment was changed over time during the simulation. The moment values ranged from 30% to 70% the length of the tooth. The limits were set as 5.0~13.5 Nmm, 4.0~11.5 Nmm, and 4.0~11.0 Nmm for the canine, the premolar, and the molar teeth, respectively. For each subject, as the magnitude of the moment increased, the rotation center of the tooth gradually shifted from the root apex to the crown. The tooth progressively changed from a distal tipping to a pure translation, and then to a distal root movement. To detect the CR position of a tooth with different penetration depths, a critical frame was selected for when the tooth was pure translated along the distal direction without tipping during the process. The exact distance between the CR and the FA point could be obtained from the value of the moment in this frame, according to Equation (2). The positioning accuracy depends on the increment of the moment value in adjacent frames. In this study, the increment was set as 0.1 Nmm to provide a precision of 0.1 mm for the CR position.

## 2.2. The Stress Distribution along PDL and the Compensation Coefficient of Orthodontic Force

Orthodontic force was applied to the tooth and transmitted to the PDL, where most OTM mechanotransduction occurred. Stresses in the PDL, rather than forces applied to the crown, were considered to be the critical regulator of bone remodeling [23], which is the biological process of OTM. The hydrostatic pressure was assumed to be the indicator of the ideal force for moving the tooth [24]. Therefore, we hypothesize that similar hydrostatic stress along the PDL generates similar bone remodeling and the tooth movement. A coefficient compensation method was used in this study to evaluate the forces for a low MSF based on this hypothesis.

It is difficult to determine the exact value of an ideal force for orthodontic tooth movement because of the individual differences in the periodontal materials and the biochemical reaction. The FEA in this study focuses on the influence of the anatomical structure of the alveolar bone. On the based of the hypothesis that a similar stress level generates similar tooth movement, we calculated the compensation coefficient of the force magnitude for the tooth penetrating into the MSF ($x > 0$ mm) so that the tooth produces a similar periodontal stress level as models with a normal periodontal anatomy ($x = 0$ mm).

The distal translation of the tooth through the MS was simulated for each subject. A force of 1 N and the corresponding moment calculated based on the M/F obtained in Section 2.1 were applied to each subject to generate a bodily movement. Models with a normal periodontal anatomy ($x = 0$) were the control group. Models with a tooth penetrating into the MSF ($x > 0$ mm) were the test groups.

The hydrostatic stress distribution along the PDL of different subjects was evaluated using FEA when the tooth moved through the sinus to different penetration depths. The

finite element model is composed of elements and nodes. The hydrostatic stress was calculated in each element.

$$\sigma_i = \frac{1}{3}(\sigma_{11} + \sigma_{22} + \sigma_{33}) \tag{3}$$

where $\sigma_{11}$, $\sigma_{22}$, and $\sigma_{33}$ are three principle stresses; $\sigma_i$ is the hydrostatic stress on the element $i$.

Therefore, the average stress on the PDL is defined as the sum of the hydrostatic stress at each element on the PDL surface divided by the number of elements as follows:

$$\sigma_{\text{ave}} = \frac{1}{N} \sum_{i=1}^{N} \sigma_i \tag{4}$$

where $\sigma_{\text{ave}}$ is the average stress on the PDL under a force of 1 N, $N$ is the number of the elements in the PDL model. It was assumed as an indicator to quantify the stress levels of each model.

The average stress of the models in the control group, expressed as $\sigma_{\text{ave}}(0)$, is the target. The average stress of the models in the test group was expressed as $\sigma_{\text{ave}}(x)$. $x$ is the penetration depth. Stress is defined as the internal force in the unit area, therefore the magnitude of orthodontic force is generally proportional to the stress. In order to produce the similar hydrostatic stress level to $\sigma_{\text{ave}}(0)$ along the PDL in the test group, the compensation coefficient of force magnitude can be expressed as:

$$R(x) = \frac{\sigma_{\text{ave}}(0)}{\sigma_{\text{ave}}(x)} \tag{5}$$

where $R(x)$ is the compensation coefficient of force magnitude for the model with $x$ mm penetration depth; $\sigma_{\text{ave}}(x)$ is the average stress along the PDL for the model with $x$ mm penetration depth under 1 N force. The compensation coefficient was used to evaluate the changes in the orthodontic forces when the teeth penetrating into the MSF.

## 3. Results

The FE simulation results present the outcome of the hydrostatic stress on the surface of the PDL and bone, and the initial displacement of the tooth under specific loads and anatomy structures. Considering that light forces are preferably used on teeth with reduced periodontal support, a 1 N force was used to compute the CR position and, in the simulation, to evaluate the stress distributions and force magnitudes.

### 3.1. CR Position and the M/F of Teeth with Different Penetration Depths

The virtual force value method was carried out by applying a constant force and a gradually increasing moment. The tooth first tipped distally, then translated, and then gradually tipped mesially. The value of the moment in the critical increment was obtained when the tooth was purely translated without tipping. Then, the CR positions were evaluated based on Equation (2). The CR positions for teeth with different penetration depths ($x$) are presented in Figure 4.

As the sinus floor sinks, the CR tends to move toward the cervical margin. The CR for an ordinary periodontal condition ($x = 0$ mm) was located approximately 69.4%, 63.4%, and 63.8% of the tooth length from crown to root apex for the canine, premolar, and molar, respectively. With the increase in the penetration depth from 0 mm to 9 mm, the location of the CR dropped linearly to 53.9%, 43.4%, and 39.9%, respectively. In the posterior teeth, the position of the impedance center even dropped to below half the length of the tooth. The CR of the molar and premolar dropped faster than that of the canine.

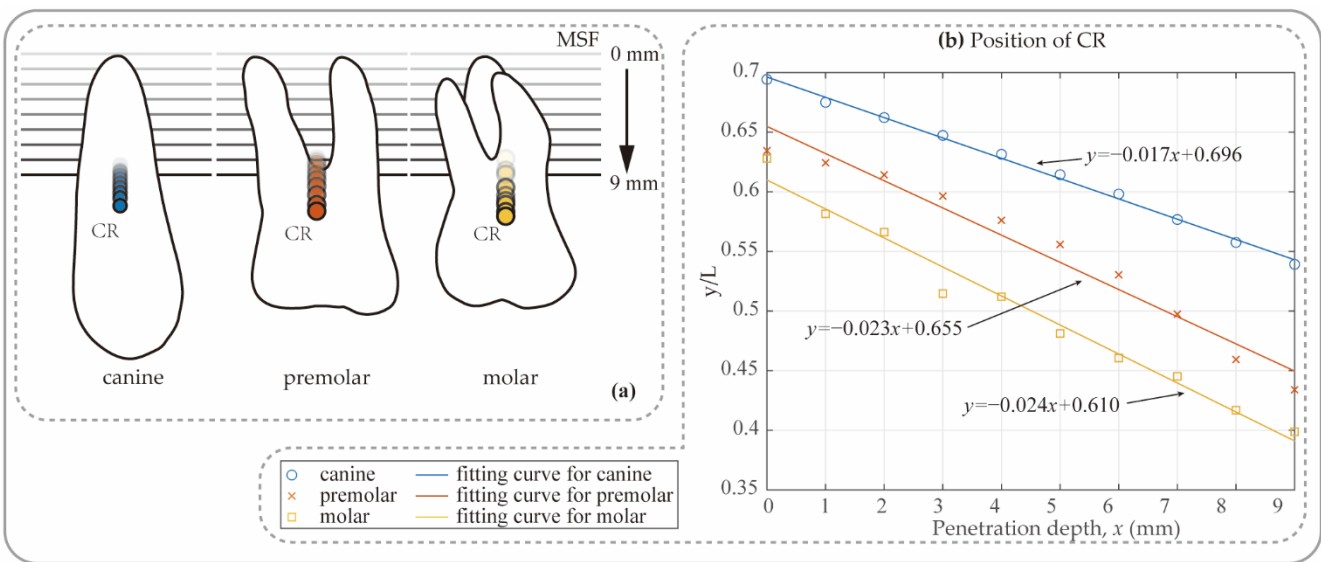

**Figure 4.** Changes of CR positions of three teeth against the penetration depths. (**a**) Schematic of the CR positions for different depths of root penetration. The blue horizontal lines present the different heights of MSFs. Colored circles with black strokes indicate the changes of CR positions because of the MSF sinking. Different transparency of the graphics represents different depths and corresponding CR positions. (**b**) The ratio of *y* to L against the penetration depth based on the simulation results. Experimental data and the linear model fitting.

The relationship between the CR position and penetration depth for the three types of teeth were described well using a linear model, as follows:

$$y/\text{L} = (\text{m}x + \text{n}) \tag{6}$$

where *y* is the distance between CR and tooth cusp, L is the length of the tooth, *x* is the penetration depth of the root, the slop m and intercept n are the parameters. The values of the parameters for each tooth are shown in Table 2.

**Table 2.** Parameters of the linear model between the CP position and penetration depth for different teeth.

| Tooth | m | n | r-squared |
|---|---|---|---|
| The Canine | −0.017 | 0.696 | 0.9965 |
| The Premolar | −0.023 | 0.655 | 0.9961 |
| The Molar | −0.024 | 0.610 | 0.9796 |

r-squared is the coefficient of determination.

Figure 4b and Table 2 show the simulation results of the CR positions against the height of the MSF. The blue lines in the graphs are the fitting curves for each tooth. The slope (m) is the descending length of the CR for every millimeter of root penetration. The descending speed of the posterior teeth was much higher than that of the canines. Anatomical structure changes have a greater impact on posterior teeth. The coefficient of determination (r-squared) values was 0.9965 for the canine, 0.9961 for the first premolar, and 0.9796 for the molar. The high r-squared fitting values indicate a good fit for the linear models.

Moreover, the orthodontic M/F for bodily movement should be equal to the distance between the bracket and the CR (H). By substituting Equation (6) into Equation (2), the relation between the M/F and the penetration depth *x* could be expressed:

$$\frac{M}{F} = (\text{m}x + \text{n})\text{L} - \text{h} \tag{7}$$

where h is the bracket height, the distance between the cusp and the bracket, which is a constant during the treatment. The CR position and the M/F in bodily movement for each type of the teeth were presented in Table 3.

**Table 3.** The CR position and the M/F for teeth with various penetration depths.

| Penetration Depth (mm) | M/F for Bodily Movement (mm) | | | CR Distance Divide by the Tooth Length, *y*/L | | |
| --- | --- | --- | --- | --- | --- | --- |
| | Canine | Premolar | Molar | Canine | Premolar | Molar |
| 0 | 11.27 | 8.00 | 8.40 | 0.69.41 | 0.6345 | 0.6279 |
| 1 | 10.82 | 7.80 | 7.50 | 0.6749 | 0.6244 | 0.5816 |
| 2 | 10.52 | 7.60 | 7.20 | 0.6621 | 0.6142 | 0.5661 |
| 3 | 10.17 | 7.25 | 6.20 | 0.6472 | 0.5964 | 0.5147 |
| 4 | 9.80 | 6.85 | 6.15 | 0.6314 | 0.5761 | 0.5121 |
| 5 | 9.40 | 6.45 | 5.55 | 0.6143 | 0.5558 | 0.4812 |
| 6 | 9.02 | 5.95 | 5.15 | 0.5980 | 0.5305 | 0.4606 |
| 7 | 8.52 | 5.30 | 4.85 | 0.5767 | 0.4975 | 0.4452 |
| 8 | 8.07 | 4.55 | 4.30 | 0.5575 | 0.4594 | 0.4169 |
| 9 | 7.64 | 4.05 | 3.95 | 0.5391 | 0.4340 | 0.3989 |

On the basis of the simulation results, an M/F of approximately 11.27 mm, 8.00 mm, and 8.40 mm can generate the bodily movement of the canine, the first premolar, and the molar with a normal periodontal anatomy, respectively. As the depth of root penetration increased, the M/F required to maintain tooth translation decreased rapidly. When the root penetrated 9 mm into the MS, the body movement of the three teeth required only 7.64 mm, 4.05 mm, and 3.95 mm, respectively.

### 3.2. Stress Distribution and Force Compensation Coefficient for Different Penetration Depths

The simulation results of the canine are exemplified in Figure 5. The simulation of teeth translation was achieved, as shown in Figure 5a, under 1 N of orthodontic force and the specific moment is shown in Table 3.

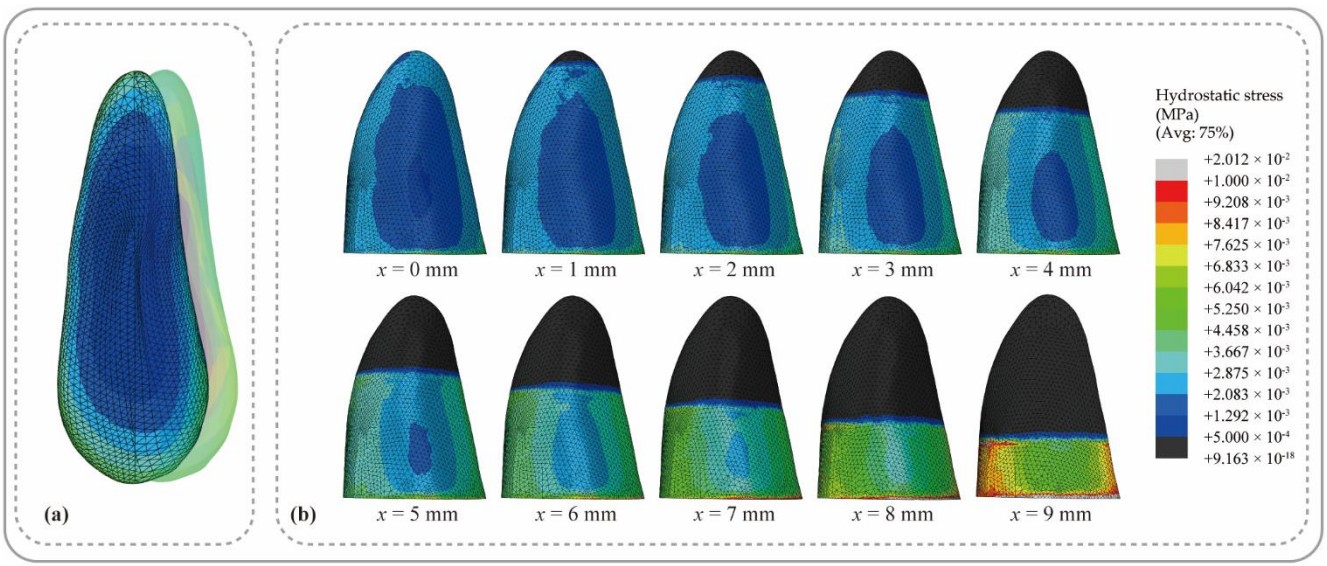

**Figure 5.** Simulation results of the stress distributions for different subjects of canine under a bodily movement. (**a**) The canine executes a bodily movement at the key increment of the simulation. (**b**) The hydrostatic stress distributions along PDL of canines with different penetration depth (*x*) during the bodily movement (labial view). PDL is the periodontal ligament.

The distal side of the PDL was under tensile stress and the mesial side was compressed. As seen in Figure 5b, the stress distribution is uniform for all subjects. However, with

the increasing penetration (from 0 mm to 9 mm), the magnitude of hydrostatic stress on the PDL gradually increased (from blue to red). The black area received nearly no stress, because the root apex protruded into the cavity and lost the support of the bone.

We quantified the stress values on the PDL for different subjects. The simulation of bodily movement was successfully achieved with the uniform hydrostatic stress distribution.

In Figure 6, the averages of the hydrostatic stresses on the PDL versus the penetration depth are shown. Under the same magnitude of the orthodontic force, the hydrostatic stress in the canine PDL was much higher than that in the premolar and molar teeth. For the tooth with a normal periodontal anatomy, the average hydrostatic stress of 2.40 KPa in the canine PDL was nearly five times larger than that in the premolar (0.57 KPa) and molar (0.54 KPa) teeth. As the depth of the root penetration increased, the stress in the PDL increased rapidly. As the penetration depth increased to 9 mm, the stress in the canine PDL increased almost threefold (7.31 KPa) as compared to when the root did not penetrate into the MS (2.40 KPa). The average stress in the premolar PDL increased fourfold from 0.56 KPa to 2.23 KPa. For the molars, the average hydrostatic stress increased from 0.54 KPa to 1.64 KPa.

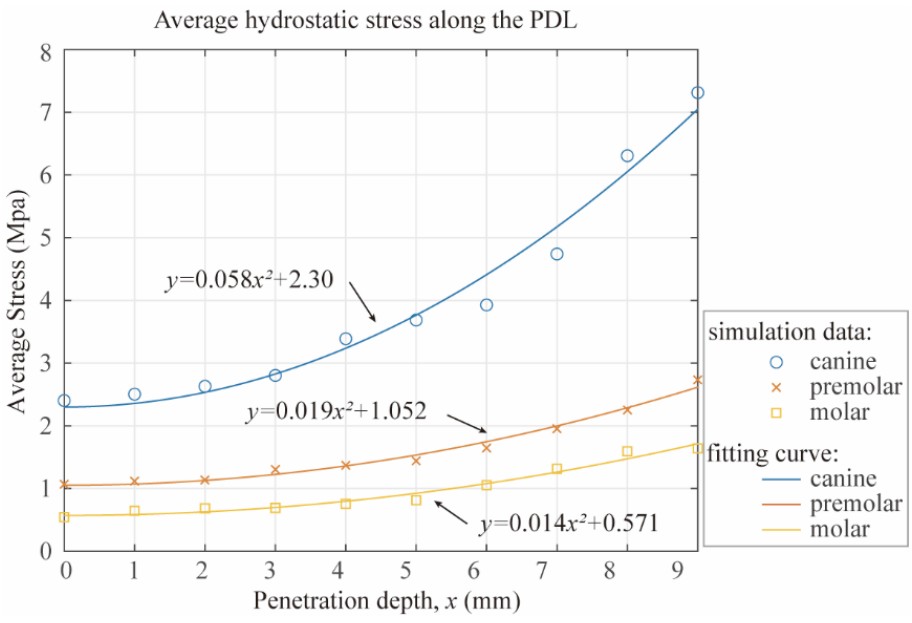

**Figure 6.** Average hydrostatic stresses along the PDL versus the penetration depths into the MSF for different teeth. Experimental data and the quadratic model fitting.

It was found that under the same orthodontic force, as the depth of the penetration increased, the average stress on PDL exhibited a quadratic increase. Similar trends can be observed in all tooth types. When the magnitude of the orthodontic force was constant, the correlation curve between average stresses and penetration depth ($x$) could be fitted by a quadratic model as follows:

$$\sigma_{\text{ave}}(x) = ax^2 + bx + c \tag{8}$$

a, b, and c are the parameters. We set b = 0, because the stresses were incremental.

Figure 6 shows stresses for each tooth versus penetration depth. The curves were fitted using the quadratic model shown in Equation (8). The fitting parameters are listed in Table 4. The quadratic model was considered a good fit because of the high r-squared values for different teeth. The quadratic term's coefficient (a) indicated the increasing speed of slopes. The stress of the canine increasing much faster than that of the premolar and the molar.

**Table 4.** Parameters of the quadratic model between the average stress and the penetration depth for different teeth.

| Tooth | a | b | c | r-squared |
|---|---|---|---|---|
| The canine | 0.05864 | 0 | 2.300 | 0.9754 |
| The premolar | 0.01931 | 0 | 1.052 | 0.9837 |
| The molar | 0.01411 | 0 | 0.571 | 0.9696 |

According to the hypothesis mentioned above, the optimum force magnitude for the tooth with a low MSF was evaluated based on the average of hydrostatic stresses along the PDL from the FE study. The reduction coefficients of the force magnitudes for different penetration depths were calculated as the reciprocal of the stress level under the same force as Equation (5). Substituting Equation (8) into Equation (5) yielded the following,

$$R(x) = \frac{c}{ax^2 + bx + c} \tag{9}$$

where $R(x)$ is the compensation coefficient of the orthodontic force magnitude and $x$ is the penetration depth. Finally, the results of stress levels and the coefficients of orthodontic forces for different subjects are shown in Table 5. The compensation coefficient was used to evaluate the changes in the orthodontic forces when the teeth penetrating into the MSF. As shown in Table 5, $R(3) = 0.81$ for the canine. It means the orthodontic force should be reduced to 81% when the tooth penetrates 3 mm deep into the MSF.

**Table 5.** Stress levels and force compensation coefficients for bodily tooth movement.

| Penetration Depth (mm) | Average Stresses along the PDL under 1 N Force ($10^{-3}$ MPa) | | | Compensation Coefficient for Orthodontic Force | | |
|---|---|---|---|---|---|---|
| | Canine | Premolar | Molar | Canine | Premolar | Molar |
| 0 | 2.4023 | 0.5674 | 0.5423 | 1.00 | 1.00 | 1.00 |
| 1 | 2.5042 | 0.6171 | 0.6458 | 0.98 | 0.98 | 0.98 |
| 2 | 2.6297 | 0.6356 | 0.6839 | 0.91 | 0.93 | 0.91 |
| 3 | 2.8029 | 0.8044 | 0.6908 | 0.81 | 0.86 | 0.82 |
| 4 | 3.3885 | 0.8695 | 0.7550 | 0.71 | 0.77 | 0.72 |
| 5 | 3.6860 | 0.9405 | 0.8124 | 0.61 | 0.69 | 0.62 |
| 6 | 3.9275 | 1.1460 | 1.0522 | 0.52 | 0.60 | 0.53 |
| 7 | 4.7395 | 1.4535 | 1.3136 | 0.44 | 0.53 | 0.45 |
| 8 | 6.3070 | 1.7520 | 1.5932 | 0.38 | 0.46 | 0.39 |
| 9 | 7.3137 | 2.2330 | 1.6391 | 0.33 | 0.40 | 0.33 |

## 4. Discussion

In this study, we attempted to establish a relationship between orthodontic forces and supporting structures influenced by the MS for different types of the teeth in a patient-specific case. The CR positions and stress distributions for teeth with different root penetration depths were evaluated. A canine, a premolar with double roots, and a molar with three roots were analyzed. The relationship between the M/F and orthodontic force magnitude against the penetration depths was established by Equations (7) and (9) based on a special case. The results shown in Tables 3 and 5 reflect the influence of the MS on the orthodontic force for bodily movement. As an example, if there are three millimeters of penetration for a canine, the 2 N force should be modified by a factor of 0.86 to 1.72 N to obtain a similar stress level to that of the 2 N force applied at a tooth that does not penetrate the MSF. Moreover, an M/F of 15.15 is needed for bodily movement of the tooth.

A virtual force method was promoted to evaluate the CR position in this study. The center of resistance was defined as the point where the applied force induced parallel movement. Many researchers determine the CR position of teeth using FE analysis. They generally adjust the acting point of the force to find the position of the CR [25]. The virtual

force method applies a force and a moment at a fixed FA point. We changed the value of the moment to obtain an equivalent effect of changing the acting point of the force. It is easier to change the value or the number of the force system than to move the force to different points. The distance between the CR and the FA point is accurate to 0.1 mm in this study. Moreover, decreasing the increment of the moment value was able to improve the accuracy of CR position.

The relationship between the CR position and the force system for a bodily movement is shown in Equation (2). It is generally accepted that the M/F is crucial for the center of rotation of tooth movement. Small miscalculations in the M/F can move the rotation center and change the type of tooth movement [26,27]. However, a more appropriate expression would be that the deviation of the M/F and the distance from the applied force to the CR (H) determine the movement pattern of the tooth. For example, fewer degrees of tipping were found for the incisors in the models with bone loss than the models with a normal bone height in response to the same force and moment [28]. This is the result of CR migration caused by bone loss. The distance H increased after the migration of the CR. Therefore, greater torque is required to tip the teeth to the same degree.

During orthodontic treatment, the moment value is limited by the anatomy and appliance geometry. Orthodontists usually control the moment by means of special bending of the arch wire, thus presetting the inclination direction of the bracket groove. The results in Tables 3 and 5 may help clinicians to calculate the orthodontic force for patients with a low MSF. In addition, a power arm attached to the bracket is another effective and convenient method to execute bodily movement [29,30]. The power arm changes the acting point and the force is applied directly through the CR. The CR position in Table 3 could help orthodontists design personalized moment arms for patients with low maxillary sinuses.

To the best of our knowledge, the average hydrostatic stress was first used to evaluate the orthodontic force. The stress distribution in the PDL offers a connection between the external forces applied to a tooth and the movement. The classic "pressure–tension" theory indicates that OTM occurs due to compression and tension within the periodontal tissues generated by orthodontic appliances [31]. A general theory of tooth movement states that in any given plane, the distance from the applied force to the center of resistance multiplied by the distance from the center of resistance to the center of rotation is a constant, which represents the distribution of the restraining stresses in the PDL [26]. Therefore, stress determines the speed and type of tooth movement. Hydrostatic stress has been used as an indicator of ideal orthodontic force in several studies [17]. They usually decide that the force is suitable to promote tooth movement based on the maximum stress not exceeding 5 kPa. In general, using the average value to evaluate the stress distribution is inaccurate. However, it could be used when the stress distributions are similar between different models as shown in Figure 5b. Therefore, the average stress could only be used to evaluate the stress level in bodily movement, in which the stress distribution is uniform.

Orthodontic treatment is a highly individualized process. The orthodontic force has a significant relationship with the material properties of the periodontal tissue, which differs between individuals. Therefore, it is difficult to provide a standard for the force value. We promoted the compensation coefficient to evaluate the changes in forces under the influence of varying root penetration depths into the MSF. The force magnitude for teeth with different penetration depths ($x > 0$ mm) is equal to the compensation factors multiplied by the force needed for the tooth with an ordinary structure ($x = 0$ mm). The compensation coefficient is reduced based on the variation in the stress level to compensate for the loss of the supporting capacity caused by the sinking of the MSF. This is consistent with various case reports in which light constant force was used to achieve the tooth movement through the MSF [32].

It should be noted that the results of the FEA are directly related to the input data, for example, the anatomical structure of the sinus floor and the tooth root. The model in this study was constructed from patient tomography images. Although the results from the special patient model herein cannot be applied directly to all patients, it is reasonable to

deduce that the linear relationship between the CR position and penetration depth and the quadratic relationship between the force magnitude and penetration depth generally exist.

Another limitation is that the MSF was considered horizontal in this study. There are various complex anatomic relationships between maxillary posterior teeth and the MSF in clinical practice [6]. A statistical study revealed that with a more vertical extension of the MSF in front of the tooth being moved, one can expect a higher degree of tipping as compared to teeth moved through a more horizontal maxillary sinus [33]. More complex conditions, including the anatomy and tooth movement types, can be analyzed in the future using a similar method.

## 5. Conclusions

The changes in the CR positions, the periodontal stresses, and the orthodontic forces when teeth penetrating into the MSF were evaluated in this study.

- The CR shifts towards the crown when the root penetrates into the MS (Table 3). As the depth of penetration increases, the CR of the canine decreases at a much lower rate than that of the premolars and molars;
- The M/F applied to the tooth should be decreased linearly with the increase in penetration depth (Table 3), because of the decrease in the CR;
- Under the same orthodontic force, the hydrostatic stresses along the PDL increase quadratically with the penetration depth (Table 5). The periodontal stress of canine teeth increases faster than that of posterior teeth;
- The force magnitude should be reduced in quadratic form with the increase in penetration depth (Table 5) to compensate for the increase in the stress in PDL.

Orthodontists should consider the individual differences between teeth and between people for the orthodontic treatment plan. The specific implications of orthodontic forces and biomechanical compression on the periodontal tissues are difficult to quantify. Therefore, FEA is mandatory in order to achieve a personal treatment plan, especially for teeth with different periodontal statuses.

**Author Contributions:** Conceptualization, X.L. and M.L.; Data curation, M.L.; Formal analysis, M.L., B.W., and B.Y.; Funding acquisition, W.T.; Investigation, X.L.; Methodology, X.L. and M.L.; Project administration, B.Y.; Resources, M.L. and J.L.; Software, X.L. and J.L.; Supervision, B.Y. and W.T.; Validation, X.L., B.W. and W.T.; Visualization, X.L.; Writing—original draft, X.L. and M.L.; Writing—review and editing, X.L., B.W., J.L., B.Y. and W.T. All authors have read and agreed to the published version of the manuscript.

**Funding:** This research was funded by the National Natural Science Foundation of China (81571005, 82071143), Key Medical Research Projects of Jiangsu Health Commission (ZDA2020003), and National Key Research and Development Plan of China (2020YFB2008102).

**Institutional Review Board Statement:** The present study was approved by the Ethics Committee of Nanjing Medical University ((2020) No.234, date of approval: 17 March 2020).

**Informed Consent Statement:** Informed consent was obtained from all subjects involved in the study.

**Data Availability Statement:** Not applicable.

**Conflicts of Interest:** The authors declare no conflict of interest.

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
