# Peer review of "Effect of the Maxillary Sinus on Tooth Movement during Orthodontics Based on Biomechanical Responses of Periodontal Ligaments"

_applsci, doi:10.3390/app12104990_

Round 1

Reviewer 1 Report

The scientific and mathematical approach is only theorical.

Nonetheless the results can give some hint to the dental movements in case of reduced PDL .

The mathematical approach seem valuable.

The english language must be revised.

Author Response

Point 1: The scientific and mathematical approach is only theorical. Nonetheless the results can give some hint to the dental movements in case of reduced PDL. The mathematical approach seem valuable.

Response 1: Thank you for your recognition of our research.

We also reorganized the math formulas.

In Method and Material:

Equation 1, 2 described the relationship between the CR position and the moment-to-force.

Equation 3 is the definition of the hydrostatic stress.

Equation 4 is a simple definition of the average hydrostatic stress.

Equations 5 describe the relationship between the orthodontic force magnitude and periodontal stresses in PDL.

In Results:

Equation 6 and Equation 8 are the multi-project models to fitting the results of CR position and the average stresses.

Equation 7 is obtained by substituting Equation 6 into Equation 2. It reflect the relationship between M/F against the penetration depth.

Similarly, equation 9 shows the relationship between the compensation coefficient of the orthodontic force magnitude and the penetration depth.

Point 2: The english language must be revised.

Response 2: We are sorry for our poor writings. We have polished the manuscript and invited a native speaker proofread it. Sentences you mentioned have been specially rephrased:

In line 19, (“The bone was manual modification for varying heights.”) was modified as “The models were manual modification to simulate different depth of root penetration.”;

Introduction line 37, (Premolar, molar, and occasionally canine roots have the opportunity to penetrate into the MS ) was replaced by “The root apices of premolar, molar, and canine may penetrate into the MS in some cases.”

We also modified some variable names in the text to make their meaning clearer:

Bone adaption was replaced by bone remodelling; Dentosinusal height was replaced by penetration depth, which indicated the depth of the root penetrated into the MS.

We hope that the revised version will be acceptable to you.

We also have requested an English editing service at https://www.mdpi.com/authors/english. However it will take a few days. We will upload the edited version in the next submission.

Reviewer 2 Report

The manuscript addresses an important matter in orthodontics, however, the poor language, grammar and structure make it difficult to follow. For example, Abstract, line 19,  (The bone was manual modification for varying heights), introduction line 37,  (Premolar, molar, and occasionally canine roots have the opportunity to penetrate into the MS ); (bone adaption [23]) and rest of the manuscript is poorly written. The abstract conclusion is inconclusive and vague. How valid equation 2 in determining CR? Authors did not run validity test. 

Author Response

Point 1: The manuscript addresses an important matter in orthodontics, however, the poor language, grammar and structure make it difficult to follow. For example, Abstract, line 19,  (The bone was manual modification for varying heights), introduction line 37,  (Premolar, molar, and occasionally canine roots have the opportunity to penetrate into the MS ); (bone adaption [23]) and rest of the manuscript is poorly written.

Response 1: We are sorry for our poor writing. We have polished the manuscript and invited a native speaker to proofread it. Sentences you mentioned have been especially rephrased:

In line 19, (“The bone was manual modification for varying heights.”) was modified as (“The models were manual modification to simulate different depth of root penetration.”);

Introduction line 37, (Premolar, molar, and occasionally canine roots have the opportunity to penetrate into the MS) was replaced by “The root apices of premolar, molar, and canine may penetrate into the MS in some cases.”

We also modified some variable names in the text to make their meaning clearer:

Bone adaption was replaced by bone remodeling; Dentosinusal height was replaced by penetration depth, which indicated the depth of the root penetrated into the MS.

We hope that the revised version will be acceptable to you.

We also have requested an English editing service at https://www.mdpi.com/authors/english. However, it will take a few days. We will upload the edited version in the next submission.

Point 2: The abstract conclusion is inconclusive and vague.

Response 2:

The conclusion in abstract was rewrite:

(During orthodontic tooth movement, the individual differences in periodontal anatomy should be considered. CR position decreases with the penetration depth, while the average hydractic stress in PDL increases rapidly. We also presented some general conclusions and reference the correction coefficients of orthodontic force and moment for tooth penetrated into the MSF. This is necessary for personalized treatment planning.)

This article mainly discusses trends and rates of change, not extreme values for individual cases. Therefore, less specific numerical values are used to describe the results.

Point 3: How valid equation 2 in determining CR? Authors did not run validity test.

Response 3: Equation 2 is the relationship between the the CR position and the moment-to-force ratio. We added the Table 2 in the section 3.1 to present the results of the distance from CR to the cusp or edge of the tooth. The location of CR has been valided in section 3.2 to generate pure translation. The constant force and a moment with a specific M/F was applied. The M/F was calculated based on Equation 2.

According to the results of the finite element analysis, bodily movement or pure translation was observed as displaced in Figure 5 (a). Therefore the CR position in Table 2, as well as the equation 2, has been valided.

Reviewer 3 Report

This paper looks at the Effect of Maxillary Sinus on Tooth Movement during Orthodontics Based on Biomechanical Responses of Periodontal Ligaments However, I think it needs a major revision to reach its full potential and usefulness. I have annotated some comments as followings:

METHODS:

The following sentences within parentheses are the aim of this study that the authors mentioned at the end of the introduction. (We aimed to assess the biomechanical response of teeth moving through MSF at different heights. The effect of the height of MSF to the bodily tooth movement was analyzed, including changes of center of resistance (CR) positions and magnitudes of stress distribution).  I could not find any answer to the aim. The answer to the aim must be clarified in the results and conclusion section clearly.

The process of building the 3D model should be described in detail as the credibility of the results depends on the accuracy of the 3D model.

On what basis do you define the material properties of each component? Any reference

What were the mechanical restrictions?

-       When measuring displacement of teeth, x-, y-, and z-axes were used. It is recommended to refer to the reference plane and reference point (0,0,0).

The unit of stress measurement should be mentioned.

You could report the Von Mises stresses in different areas or calculate the average Von Mises stress after measuring it in different areas. A table must be provided for this issue.

There are some extra formulae in the body of the manuscript that must be deleted and you cannot find them in any similar articles.

It is recommended to add some additional figures that describe (Mesial-alveolar bone), (Distal-alveolar bone), (Mesiol cusp tip), (Distal cusp tip), (Mesial root apex), (Distal root apex) of adjacent teeth to sinus before and after movement. Each image must have information regarding the amount of tooth movement, amount of stress, and …… according to finite element analysis.

To expand the discussion the following article must be cited.

The effect of Alexander, Gianelly, Roth, and MBT bracket systems on anterior retraction: a 3-dimensional finite element study Clin Oral Investig. 2020 Mar; 24(3):1351-1357. doi: 10.1007/s00784-019-03016-6. Epub 2019 Jul 28.

Author Response

Point 1: The following sentences within parentheses are the aim of this study that the authors mentioned at the end of the introduction. (We aimed to assess the biomechanical response of teeth moving through MSF at different heights. The effect of the height of MSF to the bodily tooth movement was analyzed, including changes of center of resistance (CR) positions and magnitudes of stress distribution).  I could not find any answer to the aim. The answer to the aim must be clarified in the results and conclusion section clearly.

Response 1: I realized vagueness of some sentences about the aims and the conclusion.

The aim in line 58 was modified to: (In this study, we aimed to assess the biomechanical response of teeth moving through MSF at different penetration depths. Center of resistance (CR) and periodontal stress distributions of the teeth with different depths of penetration were evaluated with finite element method (FEM). Orthodontic force and moment-to-force ratio (M/F) for bodily movements were evaluated based on the FEA results.)

The results for CR position was displaced in Figure 4. The results for stresses were displaced in Figure 5. The M/F and the orthodontic force magnitude for bodily movements or pure translation were presented in Table 2 and Table 3.

Point 2: The process of building the 3D model should be described in detail as the credibility of the results depends on the accuracy of the 3D model.

On what basis do you define the material properties of each component? Any reference

What were the mechanical restrictions?

Response 2: We have added more details about the model construction.

Line 38 (The tomography images of teeth, PDL, and alveolar bone were segmented by defining thresholds of the greyscale values in Mimics (version 19.0; Materialise, Leuven, Belgium). The model comprised tooth, 0.2 mm PDL, and manually adjusted alveolar bone. The surface models constructed in Mimics were import into Geomagic Wrap (version 2017; Geomagic, North Carolina, United States) for constructing geometry entities. After tedious model processing and repair, such as removing model noise, repairing small holes, smooth and sharp edges, etc., the geometric models were completed as displayed in Figure 3.)

About the materials of the compents, the alveolar bone and the tooth were considered to be linearly elastic materials (Liao et al., 2016). The periodontal ligament (PDL) is non-linearly. A visco-hyperelastic was developed by (Huang, Tang, Tan, & Yan, 2017) based on the results of indentation experiment.

We indicated these details in line 113.

As for the mechanical restrictions, the boundary condition is defined by fixing the top and lateral surface of the bone to reflect the natural intraoral environment during the finite element analysis.

Point 3: When measuring displacement of teeth, x-, y-, and z-axes were used. It is recommended to refer to the reference plane and reference point (0,0,0).

Response 3: I'm very sorry, we did not use the x-, y-, and z-axes to describe the displacement or stresses. There may be some misunderstanding about the characters or variables.

σx, σy, and σz were modified to σ1, σ2, and σ3. They represented the three principal stresses along three principal axes. More informations about the principal stresses could be find in the (Chandrasekharaiah & Debnath, 1994).

The variable x is the penetration depth of the root; y is the distance between the CR and the cusp or edge of the tooth. They did not represented the axes or directions. We also added a schematic (Figure 2 (a)) to presented the variables in this study.

Point 4: You could report the Von Mises stresses in different areas or calculate the average Von Mises stress after measuring it in different areas. A table must be provided for this issue.

Response 4: We used the average hydrostatic stress to evaluated the stress level in different subjects. Table 3 (Stress levels and force compensation coefficients for bodily tooth movement.) was added to provied the values of the average stress in all subjects.

Point 5: There are some extra formulae in the body of the manuscript that must be deleted and you cannot find them in any similar articles.

Response 5: We realized that some formulae are superfluous and removed equation 6, 7, and 8. The meaning and the reference of the other equations were presented here.

In Method and Material:

Equation 1, 2 described the relationship between the CR position and the moment-to-force. Similar definition could found in (Braun, Winzler, & Johnson, 1993).

Equation 3 is the definition of the hydrostatic stress (Liao et al., 2016).

Equation 4 is a simple definition of the average hydrostatic stress.

Equations 5 describe the relationship between the orthodontic force magnitude and periodontal stresses in PDL.

In Results:

Equation 6 and Equation 8 are the multi-project models to fitting the results of CR position and the average stresses.

Equation 7 is obtained by substituting Equation 6 into Equation 2. It reflect the relationship between M/F against the penetration depth.

Similarly, equation 9 shows the relationship between the compensation coefficient of the orthodontic force magnitude and the penetration depth.

Point 6: It is recommended to add some additional figures that describe (Mesial-alveolar bone), (Distal-alveolar bone), (Mesiol cusp tip), (Distal cusp tip), (Mesial root apex), (Distal root apex) of adjacent teeth to sinus before and after movement. Each image must have information regarding the amount of tooth movement, amount of stress, and …… according to finite element analysis.

Response 6: The legend for the value of displacement and stress was added in Figure 5. We analyzed the bodily movement of three teeth through the maxillary sinus in this study. We realized that just the simplified condition was discussed. This limitation is also indicated in the last paragraph in the discussion. More complexed anatomy, such as only the buccal root penetrated into the sinus, and different tooth movement types, such as tipping, should be analyzed in the future.

Point 7: To expand the discussion the following article must be cited.

The effect of Alexander, Gianelly, Roth, and MBT bracket systems on anterior retraction: a 3-dimensional finite element study Clin Oral Investig. 2020 Mar; 24(3):1351-1357. doi: 10.1007/s00784-019-03016-6. Epub 2019 Jul 28.

Response 7: We have added the discussion between the toque and the tipping movement in anterior retraction in line 340 (Cozzani et al., 2020).

Reference:

  1. Braun, S., Winzler, J., & Johnson, B. E. (1993). An analysis of orthodontic force systems applied to the dentition with diminished alveolar support. European Journal of Orthodontics, 15(1), 73–77. https://doi.org/10.1093/ejo/15.1.73
  2. Chandrasekharaiah, D. S., & Debnath, L. (1994). Continuum Mechanics. Elsevier. https://doi.org/10.1016/C2009-0-21209-8
  3. Cozzani, M., Sadri, D., Nucci, L., Jamilian, P., Pirhadirad, A. P., & Jamilian, A. (2020). The effect of Alexander, Gianelly, Roth, and MBT bracket systems on anterior retraction: a 3-dimensional finite element study. Clinical Oral Investigations, 24(3), 1351–1357. https://doi.org/10.1007/s00784-019-03016-6
  4. Huang, H., Tang, W., Tan, Q., & Yan, B. (2017). Development and parameter identification of a visco-hyperelastic model for the periodontal ligament. Journal of the Mechanical Behavior of Biomedical Materials, 68, 210–215. https://doi.org/10.1016/j.jmbbm.2017.01.035
  5. Liao, Z., Chen, J., Li, W., Darendeliler, M. A., Swain, M., & Li, Q. (2016). Biomechanical investigation into the role of the periodontal ligament in optimising orthodontic force: A finite element case study. Archives of Oral Biology, 66, 98–107. https://doi.org/10.1016/j.archoralbio.2016.02.012

Round 2

Reviewer 2 Report

Although the English language has been improved, the language and structure still require major improvements. Some sentences like this (A virtual force method was promoted to evaluate the CR position in this study) needs rephrasing and many others throughout the manuscript. Abstract conclusion is not informative to clinicians. Manuscript conclusion is hard to follow, requires major rephrasing. The manuscript needs to be re-edited, rewritten and including senior English speaking orthodontist as well as engineer to rewrite the manuscript in a presentable way.  

Author Response

Point 1: Although the English language has been improved, the language and structure still require major improvements. Some sentences like this (A virtual force method was promoted to evaluate the CR position in this study) needs rephrasing and many others throughout the manuscript. Abstract conclusion is not informative to clinicians. Manuscript conclusion is hard to follow, requires major rephrasing. The manuscript needs to be re-edited, rewritten and including senior English speaking orthodontist as well as engineer to rewrite the manuscript in a presentable way.

Response 1: To revise the manuscript, we requested an English editing service at https://www.mdpi.com/authors/english. Edits are shown as tracked changes in the second revised version. The sentences you mentioned was specially rephrased:

(A virtual force value method was used in this study to evaluate the CR position of the specific tooth based on the relationship between the CR position and the M/F for bodily movement in Equation 1.) (line 152)

I realized the results of the study must be more specific. This article mainly discusses the changes in the CR position, the stress, and the orthodontic force. Therefore, we added two tables (Table 4 and 6) to present the slop and other parameters of the fitting model (Equation 7 and Equation 9).

Since the abstract must be no more than 200 words, its hard to present all the results about the moment-to-force values (Table 4) and orthodontic force magnitudes (Table 6) for each teeth. Therefore, in the abstract conclusion, we just point out what we can expect from this study (In this paper, we present the correction coefficients of the orthodontic force and moment for a tooth penetrating into the MSF).

In manuscript conclusion, we concluded the trends of the changes in the CR position, the stress, and the orthodontic force for teeth penetrating into the MSF. We added a general sentence to summarized the conclusion in line 409: (The changes in the CR positions, the periodontal stresses, and the orthodontic forces when teeth penetrating into the MSF were evaluated in this study.) This may be helped in understanding the next four sub-arguments in the conclusion.

Moreover, maybe the results of the “compensation coefficient” is hard to follow.

It’s difficult to determine the exact value of the force for moving tooth because of the individual differences as we discussed in line 180. Therefore, the compensation coefficient (R(x)) was used to evaluated the changes in the orthodontic forces when the teeth penetrating into the MSF. We added the interpretation in line 324. (As shown in Table 5, R(3)=0.81 for the canine. It means the orthodontic force should be reduced to 81% when the tooth penetrates 3mm deep into the MSF.)

Reviewer 3 Report

The article has improved significantly and it is publishable.

Author Response

Point 1: The article has improved significantly and it is publishable.

Response 1: Thanks for the encouraging comments. Moreover, we submitted the manuscript to MDPI for English editing. Edits are shown as tracked changes in the second revised version.
